# Collaborative Planning of Community Charging Facilities and Distribution Networks

Xiao-Hong Diao [1], Jing Zhang [1,*], Rui-Yu Wang [2], Jiang-Wei Jia [3,*], Zhi-Liang Chang [3], Bin Li [1] and Xuan Zhao [1]

[1]  Beijing Laboratory, Beijing Engineering Technology Research Center of Electric Vehicle Charging/Battery Swap, China Electric Power Research Institute, Beijing 100192, China; jjw2jrq@163.com (X.-H.D.)
[2]  Nanyang Feilong Power Supply Service Co., Ltd., Xinye County Branch, Nanyang 473500, China; 13752736409@163.com
[3]  School of Electrical Engineering, Tiangong University, Tianjin 300387, China; czl2lk@163.com
*   Correspondence: opkl_5606@163.com (J.Z.); 15530041917@163.com (J.-W.J.)

**Abstract:** The construction of community charging facilities and supporting distribution networks based on the predicted results of electric vehicle (EV) charging power in saturation year has resulted in a large initial idleness of the distribution network and a serious waste of assets. To solve this problem, this paper proposes a collaborative planning method for urban community charging facilities and distribution networks. First, based on the load density method and occupancy rate to predict the base electricity load in the community, the Bass model and charging probability are used to predict the community's electric vehicle charging load. Taking the minimum annual construction and operation costs of the community distribution network as the objective function, the power supply topology of the distribution network for a new community is optimized by using Prim and single-parent genetic algorithms. Finally, the proposed scheme is verified by using the actual community data of a certain city in China as an analysis example, and the scheme of one-time planning of the distribution network and yearly construction of charging facilities is given.

**Keywords:** collaborative planning; charging facility; distribution network; electric vehicle; charging load forecasting; community





## 1. Introduction

With a series of policies to promote the development of the electric vehicle industry, the number of electric vehicles and basic charging facilities in China is growing rapidly. Large-scale EV charging loads are connected to the distribution network of the community in a disorderly manner, which makes the existing buildings, line corridors, and distribution facilities limited, and makes the existing community face problems such as difficulties in the transformation of the distribution network or lack of conditions for the transformation of distribution network capacity. However, when charging facilities are planned and constructed in community areas, supporting distribution networks are often constructed according to the forecast of expected saturated charging power. As a result, the initial idle resources of the distribution network are large, the number of charging facilities does not match the scale of electric vehicles, and the layout is unreasonable. When a large number of electric vehicles are charged centrally at a certain time or place, the charging load agglomeration effect will be formed, resulting in an extreme peak of the load of the distribution network. As a result, the node voltage of the distribution network will exceed the limit and the network loss will increase, thus endangering the normal operation of capacitors, transformers and other equipment and damaging the service life of the equipment. Therefore, the safe operation and reasonable planning of the distribution network are facing new challenges [1–3].

For the typical problems facing the rapid development of electric vehicles at present, it is necessary to study the charging load prediction and charging facility planning in the

typical scenes of newly built communities. In view of the prediction of electric vehicle ownership, the elastic coefficient method and the thousand-people ownership method are adopted in references [4,5] to predict electric vehicle ownership. However, due to the rapid development of the Chinese economy to high-quality development, the actual data differ greatly from the predicted data in recent years. According to the authors in [6], the ownership prediction model of new energy vehicles under the control of multiple factors was constructed by considering macro factors such as policy and economy and micro factors such as vehicle price, respectively.

Using the Bass model [7] to forecast the number of electric vehicles, [8] conducted a detailed analysis of the impact of the model's three main parameters (imitation coefficient, innovation coefficient, and market potential) on the accuracy of the prediction. The authors in [9] analyze the influence of social network forms and EV business models on the promotion of EVs by network control theory and simulate the diffusion by the Bass model.

Ev charging load prediction is the key point of connecting existing data and planning schemes. By forecasting the charging load of electric vehicles, the existing statistical data such as the number of electric vehicles, users' travel characteristics and users' charging habits can be converted into the forecasting of charging vehicle number and charging load required by the planning of charging facilities, which is conducive to improving the accuracy of planning schemes and reducing unnecessary investment and asset waste. The authors in [10] fitted the probability distribution of each characteristic quantity in the travel chain (travel time, travel time, mileage, etc.), and proposed a random charging demand simulation method based on the travel chain. The authors in [11] established the spatial-temporal distribution model of EV charging load through multi-agent simulation. The authors in [12] reveal the spatial load changes in different development stages of residential districts through cluster analysis of data such as maximum load and power consumption of residential districts.

A sound network of electric vehicle charging facilities and power supply services is the basis for the rapid development of the electric vehicle industry. The authors in [13] take the minimum user charging cost as the objective function and use a genetic algorithm to solve the location planning of charging facilities for a given number of charging piles and carry out the location determination capacity according to the charging cost of different users. The authors in [14] mainly consider reducing power grid operation and user charging costs and use travel probability to calculate charging demand, so as to determine the location and capacity of fast charging stations. The authors in [15], the road segment propagation model LTM was used to simulate the temporal and spatial distribution of traffic flow, determine the time for EVs to arrive at charging stations in service areas of expressways, and calculate the queuing time and charging load demand of charging stations through the M/M/S EV queuing model, which has guiding significance for the number planning of charging piles. An EV charging prediction model was established by the authors in [16], considering various factors related to charging that simulated the current and future EV charging load distribution characteristics; they carried out scientific planning and construction of charging stations and analyzed the payback period of charging facilities investment. The authors in [17] point out that the ratio of charging modes in charging stations should be planned with full consideration of the charging preferences of user groups facing charging facilities, and the ratio of fast and slow charging piles should be adjusted according to the load distribution characteristics of electric vehicles in the region, so as to minimize construction and maintenance costs. The authors in [18], collaborative planning of charging facilities and distribution systems is mainly considered in order to reduce the negative impact on the distribution network and minimize the total investment and operating cost through the two-layer optimization model. By combining charging facilities for different charging types and user selection criteria, [19] seeks to identify suitable locations for deploying community Electric Vehicle charging points using a Geographic Information System (GIS)-based approach.

The above references provide a reference for EV charging load prediction and charging facility planning in urban newly built communities. In this paper, based on the load density method and the occupancy rate of the new community, the base electricity load of the new community is obtained and determines the growth of EV ownership in urban areas by comparing the EV ownership in cities with that in the whole country. The prediction model of electric vehicle ownership in a community is established. Secondly, the travel characteristics of EV users in urban communities are analyzed based on statistical data, and the charging load prediction model of EVs in a community is established based on the Monte Carlo method. Then, the power supply topology optimization model of the distribution network in the newly built community is established by taking the minimum average annual cost of the distribution network in the community as the objective function. Considering that the topology is radial and satisfies the planned economy, this paper adopts the Prim minimum spanning tree algorithm to generate the initial topology of the distribution network and uses the single-parent genetic algorithm to find the optimal topology under different topologies. Finally, the method is verified by simulation with a practical case, and the plan of distribution network planning and charging facilities construction is given year by year.

## 2. New Community Collaborative Optimization Construction Method Framework

The new community can be configured according to the predicted results due to the topology of the distribution network, the power supply line and the capacity of the distribution transformer, so as to reduce the large-scale maintenance problems caused by the insufficient capacity of the line. When the location of the load node in the newly built community is known, the acceptance capacity of the distribution network in the target year is determined by predicting the base load and electric vehicle charging load of the community. The topology structure of the distribution network and the construction scale of charging facilities are determined by the optimization algorithm. Figure 1 shows the collaborative optimization construction ideas of the newly built community.

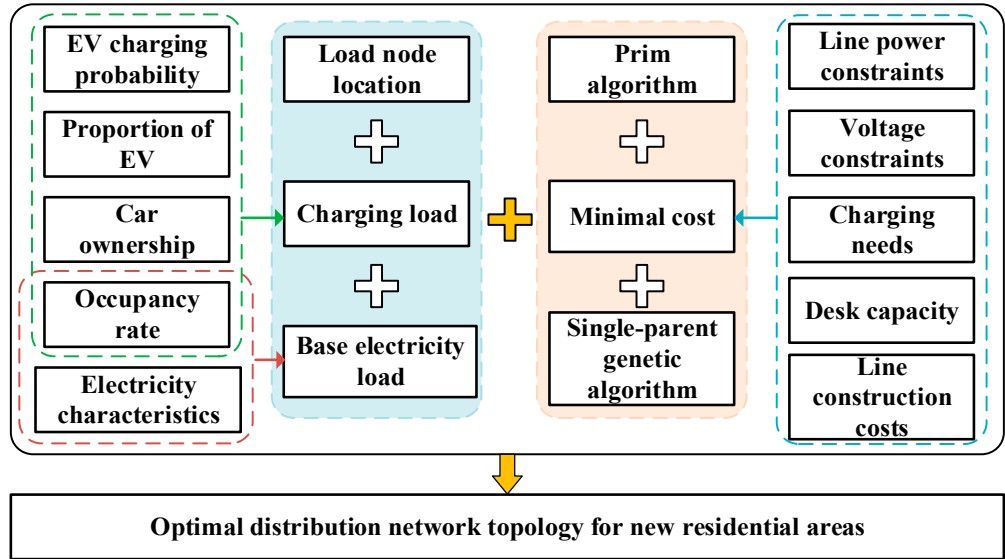

**Figure 1.** Collaborative optimization construction of new communities.

(1) Calculate the basic electricity load of the new community in the target year by analyzing the occupancy rate, planned number of households and residential electricity characteristics, so as to plan the platform capacity of each load node;

(2) Carrying out data analysis on the number of cars owned by 1000 people and the proportion of electric vehicles in the city where the newly built community is located, combined with the prediction results of the charging probability of electric vehicles,

the charging load of electric vehicles in the newly built communities in the target year can be predicted;

(3) Take residential buildings as base load nodes, calibrate base load locations in newly built communities, calculate load density, and determine the location of EV charging nodes and 10 kV outlet nodes;

(4) Considering constraints such as line power, voltage amplitude, charging demand and platform capacity of distribution network in newly built communities, an economic model of collaborative optimization construction between electric vehicle charging facilities and distribution network in newly built communities is established. Combining the Prim algorithm and the single-parent genetic algorithm, the topology of the distribution network grid, the line type of the feeder, etc., are determined, and the construction of the charging facilities is planned.

## 3. Forecasting Base Load and Charging Load in New Communities

### 3.1. Prediction of Basic Electricity Load

The urban community residential building can be regarded as a base load node, and its base load prediction can use the load density method. The target annual base load of the new community is:

$$P_{base}(i) = \frac{A_{house}(i) D_H(i) K_{Area}(i) \lambda}{1000} \quad i \in N_{base} \tag{1}$$

where: $P_{base}(i)$ is the base load of the $i$-th base load node in the newly built communities in the target year (unit: kW); $A_{house}(i)$ is the residential housing area of the $i$-th base load node, and the unit is m$^2$; $D_H(i)$ is the load density index of the $i$-th base load node, and the unit is W/m$^2$; $K_{Area}(i)$ is the area requirement coefficient of the $i$-th base load node; $\lambda$ is the occupancy rate of communities in the target year; $N_{base}$ is the number of base load nodes in the community.

### 3.2. EV Charging Load Prediction

Ev charging load is mainly determined by EV quantity and EV charging probability in communities. Among them, the growth of EV ownership in communities mainly depends on the number of thousands of urban cars, resident occupancy rate and EV growth trend, and EV charging probability mainly depends on the EV owners' travel habits. The statistics of the number of cars owned by 1000 people in cities include the number of electric vehicles and the number of fuel vehicles. The number of electric vehicles in communities can be expressed as:

$$Q_{EV}(j) = O(j) N_p(j) \lambda(j) \varphi(j) \tag{2}$$

where: $Q_{EV}(j)$ is the number of electric vehicles in communities in the $j$-year, unit: vehicle; $O(j)$ is the number of cars per thousand people in the $j$-year of the city, and the unit is: cars per thousand people; $N_p(j)$ is the number of community households in the $j$-year, unit: 1000; $\varphi(j)$ is the proportion of electric vehicles in civil vehicles in the $j$-year of the city.

3.2.1. Prediction of Electric Vehicle Ownership in Communities

The private car ownership and growth rate data of a city from 2011 to 2019 are shown in Table A1 of Appendix A [20].

Based on the data on private car ownership and growth rate in Table A1 of Appendix A, the fitting tool in MATLAB is used to fit the growth rate curve of private car ownership in the city, as shown in Figure 2.

The fitting curve function of the growth rate of private car ownership in this city can be expressed as:

$$f_{growth}(x) = a e^{-\left(\frac{t-b}{c}\right)^2} \tag{3}$$

where, $a$ = 0.4652, $b$ = 2003, $c$ = 12.51.

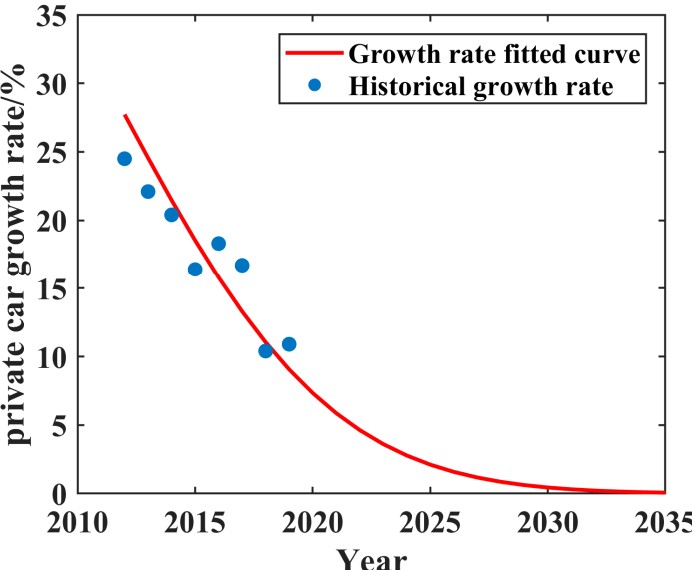

**Figure 2.** Fitting curve of growth rate of private vehicle ownership.

According to the Bass model in the durable goods diffusion model, the medium and long-term ownership of electric vehicles in the community is predicted, and the population of electric vehicles in the community in the next few years is obtained. The calculation formula is as follows:

$$f(t) = Mp[1 - \frac{F(t)}{M}] + qF(t)[1 - \frac{F(t)}{M}] \tag{4}$$

where $F(t)$ is the number of electric vehicles at time $t$; $f(t)$ is the number of newly added electric vehicles at time t; $M$ is the maximum acceptance of electric vehicle users; $p$ is the external influence coefficient, the empirical range is 0.01~0.03; $q$ is the internal influence coefficient, and the empirical range is 0.3~0.7 [21].

3.2.2. EV Charging Probability Prediction in Communities

(1)　Starting time of electric vehicle charging

The travel characteristics of EV users are closely related to the charging start time of the EV, which is usually charged at the end of a day's journey. Therefore, the charging start time of EVs in communities is the end time of a day's journey for EVs in communities. Therefore, this paper replaces the travel characteristics of electric vehicles in communities with the travel characteristics of private fuel vehicles in ordinary communities. The probability distribution at the end of working days and weekend travel for private cars in this paper is the same as the statistical data collected by the NHTS (National Household Travel Survey) of the US Department of Transportation in 2017 [22,23].

(2)　Charging start SOC

The probability density distribution of single trip mileage of electric vehicles in communities is lognormal [24,25], and its probability density function can be expressed as:

$$f_D(d) = \frac{1}{d\sigma_D\sqrt{2\pi}}e^{-\frac{(\ln d - \mu_D)^2}{2\sigma_D^2}} \tag{5}$$

where $d$ is the number of kilometers per trip of electric vehicles in communities; $\mu_D$ is the average of the logarithm of the mileage of a single trip of electric vehicles; $\sigma_D$ is the standard deviation of the logarithmic mileage of a single trip of an electric vehicle. In this paper, the single-trip driving distance of electric vehicles in communities on working

days is set as $\mu_D = 11.4$, $\sigma_D = 4.88$ km; the mileage of single trips of electric vehicles in communities on weekends $\mu_D = 13.2$ km, $\sigma_D = 5.23$ km [26].

Starting SOC for EV charging in communities:

$$SOC_0 = (1 - \sum_{k=1}^{n} \frac{d_k}{d_{full}}) \times 100\% \tag{6}$$

where $SOC_0$ is the initial SOC of EV charging; $d_k$ is the mileage of the $k$ trip of the electric vehicle; $d_{full}$ represents the maximum number of miles an electric car can travel on a full battery charge.

(3)　Charging time

The formula for calculating the charging time of electric vehicles in communities is:

$$t_c = \frac{(1 - SOC_0)E}{P_c} \tag{7}$$

where $t_c$ is the charging time of the electric vehicle (unit: h); $SOC_0$ is the initial SOC of charging for electric vehicles; $P_c$ refers to the charging power of electric vehicle charging facilities (unit kW); $E$ is the battery capacity of electric vehicles (kW·h).

(4)　Monte Carlo simulation charging probability prediction

Based on the number of EVs in the community in the target year and the established mathematical models of the starting time of EV charging, the starting SOC of EV charging and the charging time of EV charging, the Monte Carlo simulation method is used to solve the charging probability of EV in working days and weekends, respectively. The number of electric vehicles in the charging state at each moment is calculated by combining the medium and long term forecast results of electric vehicle ownership in communities with the charging probability. Furthermore, the charging load of EVs in communities can be obtained by combining the number of EVs in the charging state and EV charging power at every moment [27].

## 4. Mathematical Model of Collaborative Optimization Construction of Community
### 4.1. Objective Function

The main cost of a distribution network comes from construction costs and network loss costs. Taking the minimum annual comprehensive cost as the objective function, the objective function of distribution network planning for newly built communities is:

$$F = I\left[\frac{r_0(1 + r_0)^y}{(1 + r_0)^y - 1}\right] + L \tag{8}$$

where: $F$ is the annual comprehensive cost of the newly built residential area, and the unit is: ten thousand yuan/year; $I$ is the cost of line construction in newly built communities, unit: ten thousand yuan; $r_0$ is the discount rate; $y$ is the economic service life of the line, and the unit is: year; $L$ is the cost of line loss in newly built communities, unit: ten thousand yuan/year.

Different lines are adopted in the construction of lines for newly built communities to reduce costs. The construction costs of lines for newly built communities are as follows:

$$I = \sum_{k \in N_k} (l_k f_{D_k}) \tag{9}$$

where: $l_k$ is the length of line $k$, and the unit is km; $f_{Dk}$ is the construction cost when the cross-sectional area of line $k$ is $D_k$, and the unit is yuan/km; $N_k$ indicates a new line collection.

The network loss cost of lines in newly built communities is:

$$L = \sum_{k \in N_k} (\alpha \frac{P_k^2 + Q_k^2}{U^2} g_{D_k} l_k \tau) \tag{10}$$

In the formula, $\alpha$ is the electricity price, and the unit is yuan/kWh; $P_k$ is the active power flowing on line $k$, and the unit is kW; $Q_k$ is the reactive power flowing on line $k$, and the unit is kVar; $U$ is voltage, unit: kV; $g_{D_k}$ is the resistance per unit length when the cross-sectional area of line $k$ is $D_k$. The unit is $\Omega$/km. $\tau$ is the maximum number of hours of load utilization per year, expressed in hours.

*4.2. Constraints*

(1) Distribution network line capacity constraints

During the operation of the distribution network, the power flowing through the planned lines should be smaller than the planned line capacity.

$$P_k \leq P_k^{\max} \tag{11}$$

where: $P_k^{\max}$ is the maximum capacity of the $k$ line of the distribution network, and the unit is kVA.

(2) Distribution network node voltage constraints

The radial topology of the distribution network tends to cause the terminal node voltage to drop seriously, which cannot meet the normal demand for electricity. The voltage constraint of distribution network nodes in newly built communities is:

$$U_{\min} \leq U_n \leq U_{\max} \tag{12}$$

where: $U_{\min}$ is the lower limit of the voltage of distribution network nodes; $U_{\max}$ is the upper voltage limit of distribution network nodes; $U_n$ is the lower limit of the nodal voltage of the NTH node in the distribution network, and is per unit value.

(3) Constraints on charging demand

Multiple EV charging nodes are planned to serve EVs in the distribution network of newly built communities, and their node capacity should meet the EV charging load requirements under their respective nodes. The charging demand constraint of the communities' distribution network is:

$$P_{char}(i) \leq C_{char}(i) \tag{13}$$

where, $P_{char}(i)$ is the charging load of the $i$-th EV charging node in the distribution network, and the unit is kW; $C_{char}(i)$ is the variable capacity of the $i$-th EV charging node in the distribution network, and the unit is kVA.

(4) Capacity constraints of distribution network station area

The load of each node in the distribution network should not exceed the upper limit of its node distribution capacity, and a certain margin should be left for the base load in the later period. The station area capacity constraint of the residential distribution network is:

$$P_{norm}(i) \leq C_{norm}(i) \tag{14}$$

In the formula, $P_{norm}(i)$ is the $i$-th resident base node load of the distribution network, and the unit is kW; $C_{norm}(i)$ is the distribution capacity of the $i$-th communities base node in the distribution network, and the unit is kVA.

## 5. Prim and Single-Parent Genetic Algorithm

The load nodes of the distribution network in communities are located near residential buildings in communities. Due to the factors of line structure, distribution capacity and

land area, most charging facilities in old communities are connected to base load nodes, which is not easy to maintain, transform and upgrade charging nodes, and easily affect residents' lives. Therefore, electric vehicle charging load can be formed into a single power supply node during distribution network planning in newly built communities, so as to facilitate maintenance and upgrading without affecting residential power consumption in the later period.

### 5.1. Prime Algorithm

Distribution network topology is generally radial, and the cost of line laying is high. Considering the economy of topology planning and radial topology, the Prim minimum spanning tree algorithm is used to generate the initial topology of the distribution network. The tree formed by the edge subset searched by the Prim algorithm not only includes all vertices in the connected graph but also has the smallest sum of weights of all edges [28]. The process of the Prim algorithm is shown in Figure 3. The main steps are as follows:

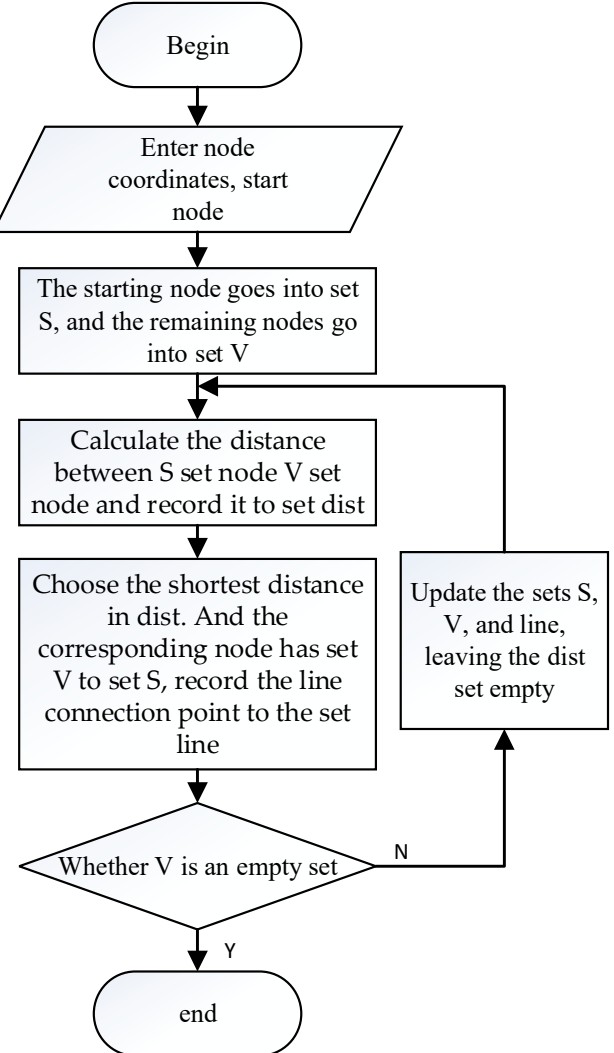

**Figure 3.** Prim algorithm flow chart.

Step 1: Enter the coordinate position of each load node of the distribution network and the start node of the network topology;

Step 2: Set up set S storing connected load nodes, set V storing unconnected load nodes, set dist storing connecting line distance between nodes of set S and nodes of set V, and set line storing selected lines. See Figure 4 for the data storage methods of each set.

dist set

| Root node (S) | child node (V) | Two-node distance |
|---|---|---|
| Root node (S) | child node (V) | Two-node distance |

...

| Root node (S) | child node (V) | Two-node distance |
|---|---|---|

**Figure 4.** Dist collection data storage mode.

Step 3: Update sets S and V, empty set dist, calculate the line distance between each node of set S and each node of set V and store it in set dist;

Step 4: Select the shortest line distance in set dist as the connection line under the topology, add the selected line to the set line, and add the selected load node from set V to set S;

Step 5: Judge whether set V is an empty set, if so, end and output the result; Otherwise, return to Step 3.

Based on the above steps, the shortest topology of the community distribution network can be realized under the constraint conditions, and the solution speed of the distribution network topology can be accelerated.

*5.2. Single-Parent Genetic Algorithm*

The Parthenogenetic Algorithm is similar to the traditional Genetic Algorithm, and it is also a population algorithm using a random search method. The basic principle is to use genetic operators to act on the parent population in order to produce a more adaptable offspring population and iteratively perform this operation to achieve population evolution until the termination condition. Compared with the traditional genetic algorithm, the advantage of a single-parent genetic algorithm is that it uses its own operator for shifting or redistribution inheritance, so as to avoid the duplication or deletion of topological lines caused by the cross-inheritance of two operators [29].

The encoding mechanism has a great influence on the optimization performance of genetic algorithms. The adoption of an appropriate encoding method can not only make the problem expression more intuitive and solve more convenient, so that some originally difficult problems can be solved easily, but also can effectively compress the optimization space and improve the solving efficiency to a certain extent. On the one hand, the single-parent genetic algorithm encoded by tree structure is combined with the spanning tree method to generate the initial population, and the generated initial distribution network naturally radiates. On the other hand, due to the inherent characteristics of the encoding mode and genetic operator, the optimization process does not need decoding and the possibility of scheme connectivity and radiation being damaged in the optimization process is eliminated, which can simplify the operation process [30].

In order to avoid the traditional genetic algorithm breaking the radial constraint of distribution network due to operator crossing, the single parent genetic algorithm adopts



displacement and redistribution operation to carry out population iteration, so as to realize population evolution.

The fitness function adopts the economic objective of cooperative optimization construction, namely construction cost and network loss annual cost. For fitness functions that do not meet constraint conditions under different topologies, this project adopts a penalty function to punish. The specific steps of the single-parent genetic algorithm are shown in Figure 5.

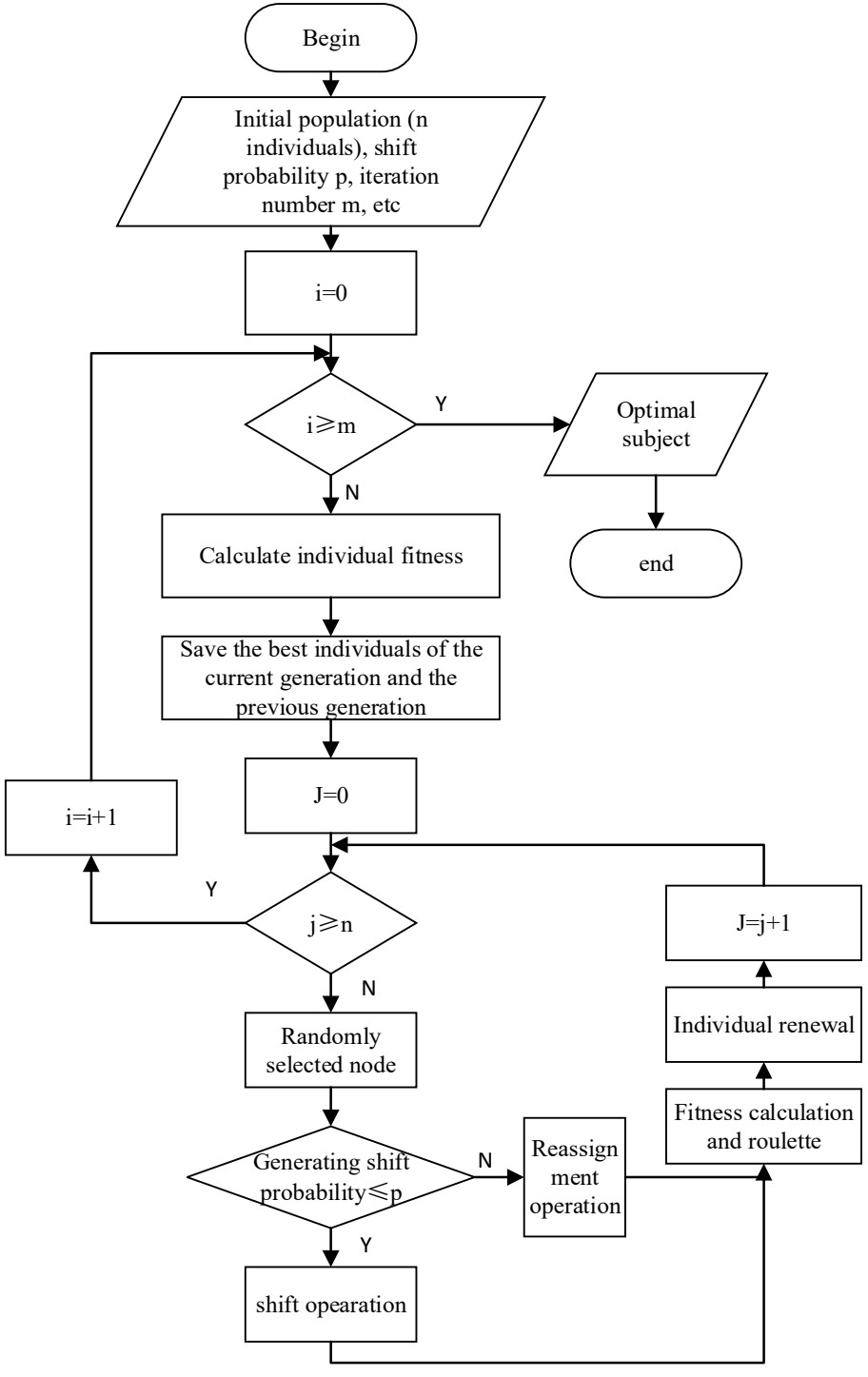

**Figure 5.** Single—parent genetic algorithm flow chart.

## 6. Analysis of Examples

This section takes a community in a Chinese city as an example. It plans to build 13 residential buildings, with a building area of 177,300 m² and 1298 households and adopts 1:1 parking space allocation of underground parking lots. Two types of cables, YJV22-3*150 and YJV22-3*300, are used to construct the lines in the community area. The related parameters and construction costs are shown in Table A2 of Appendix A.

The planning and construction scheme of each unit building in the community is shown in Figure 6. The planned number and building area are shown in Table 1. The nodal location of the unit building is shown in Figure 7.

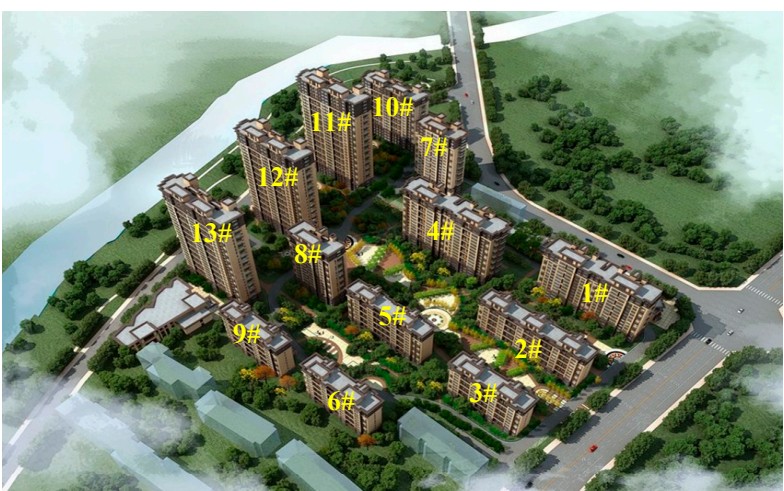

**Figure 6.** Planning rendering of the community area.

**Table 1.** Planning and construction of residential buildings in newly built communities.

| Number | Number of Households | Building Area/m² | Number | Number of Households | Building Area/m² |
|---|---|---|---|---|---|
| 1# | 120 | 15,600 | 8# | 72 | 7920 |
| 2# | 66 | 8580 | 9# | 44 | 5720 |
| 3# | 44 | 5720 | 10# | 120 | 15,600 |
| 4# | 144 | 18,720 | 11# | 180 | 27,000 |
| 5# | 44 | 6600 | 12# | 174 | 26,100 |
| 6# | 44 | 5720 | 13# | 174 | 26,100 |
| 7# | 72 | 7920 | | | |

According to relevant studies [31], the occupancy rate of newly built communities shows a rapidly rising trend after the completion and delivery, reaching more than 90% in the four years after the completion, and then begins to grow slowly and gradually approaches 1. The specific growth is shown in Figure 8. By predicting the number of electric vehicles in the city and the charging probability of electric vehicles, the conventional electricity load and charging load of electric vehicles in the newly built communities can be predicted by combining Equations (1) to (2).

### 6.1. Forecast of Electric Vehicle Ownership in Newly Built Communities

Both new communities and old communities belong to urban communities. Based on the forecast results of the number of civilian cars owned by 1000 people in an urban community and the proportion of electric vehicles, and starting from the fact that the proportion of electric vehicles in a certain city in 2021 is 1.35 times the proportion of electric vehicles in the country, and combined with the characteristics and fitting parameters of the Bass model, we set the proportion of electric vehicles to 1.35 times the national proportion

of electric vehicles before reaching 45% of civilian vehicles (set the saturation value of electric vehicles to 90% of civilian vehicles for parameter fitting); the proportion of electric vehicles and the growth rate after the level reaches 45% of civilian vehicles is the reverse value of the previous growth rate. Combined with the study on the occupancy rate of new urban communities in Figure 8, electric vehicle ownership in new communities is predicted, and the forecast results are shown in Figure 9.

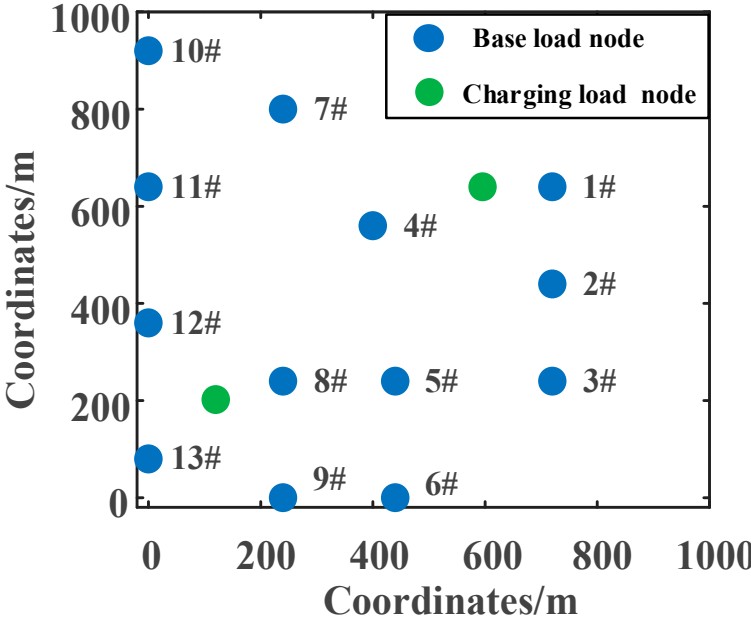

**Figure 7.** Unit building nodalization location map.

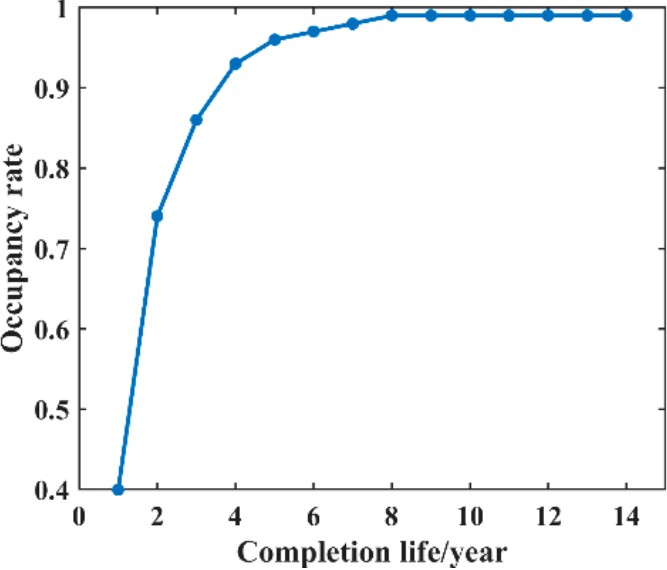

**Figure 8.** Change in occupancy rate of newly built communities.

As shown in Figure 9, due to the rapid rise of the occupancy rate, the number of civilian vehicles in the newly built communities shows a rapidly rising trend in the years before occupancy. However, due to the year-by-year decline in the growth rate of urban civilian vehicles, the number of community civilian vehicles also tends to be saturated when the occupancy rate is close to saturation. In addition, due to the rapid growth of the proportion of electric vehicles, their ownership shows a linear growth trend and will reach 50% of the saturation value around 2030 and approach the saturation value around 2040.

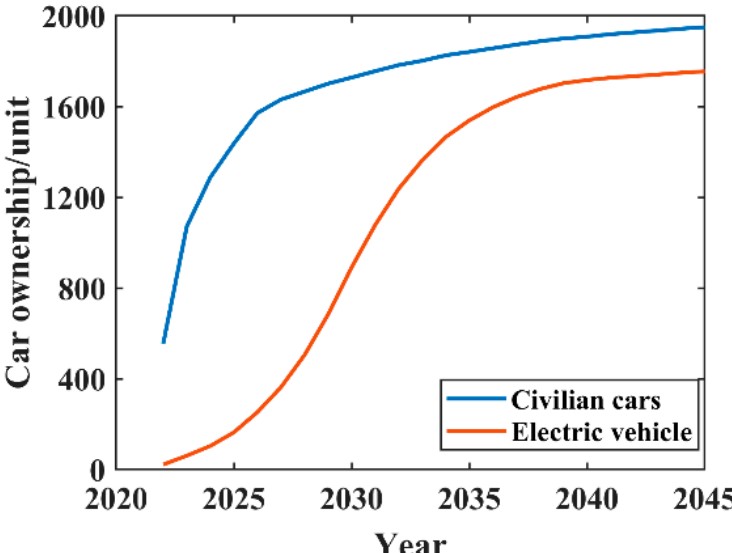

**Figure 9.** Forecast of vehicles ownership in community areas.

According to prediction results, this paper takes 2040 as the planned saturation year and predicts the residents' base load and EV charging load from the completion of the community area to the planned saturation year, respectively.

*6.2. Prediction of Base Load and Electric Vehicle Charging Load in Newly Built Communities*

The load density was set at 40 W/m², the coefficient of area demand was 0.17, and the annual growth rate of the base load was 1%. The characteristics of residential conventional power load were shown in Figure 10. It is easier for this urban community to reach the saturation value of accepting the capacity of electric vehicles on summer working days. Therefore, summer working days in Figure 10 are used to calculate the peak base load of each node in this residential area, and the required capacity of each node is planned once. The results are shown in Table A3 of Appendix A. Taking 2022, 2025, 2030, 2035 and 2040 as the years to be planned, the charging power of electric vehicles is set as 7 kW, and the charging load of electric vehicles in the newly built communities is predicted. The results are shown in Figure 11.

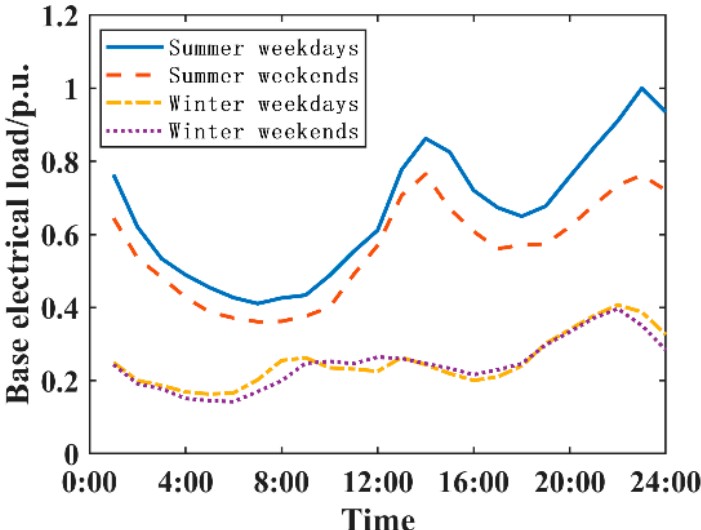

**Figure 10.** Typical daily base electric load characteristics in communities.

As can be seen from Figure 11, due to the rapid growth of the occupancy rate in the newly built communities, the residential base electricity load presents a rapid growth

trend and increases slowly after the occupancy rate is saturated. On the other hand, the EV charging load in communities increases rapidly after the construction of communities due to the growth of residents' occupancy rate and the increase of the proportion of EVs. After the proportion of EVs is saturated, the EV charging load increases slowly due to the increase in the number of thousand vehicles in the city. To sum up, the base electricity load of the newly built residential area will grow more rapidly than that of the charging load of electric vehicles. When the occupancy rate approaches saturation, the charging load of electric vehicles will be in a rapid growth stage. Therefore, detailed planning of the charging facilities and power distribution facilities of the newly built community should be made in the early stage of residential area construction to avoid idle resources and waste.

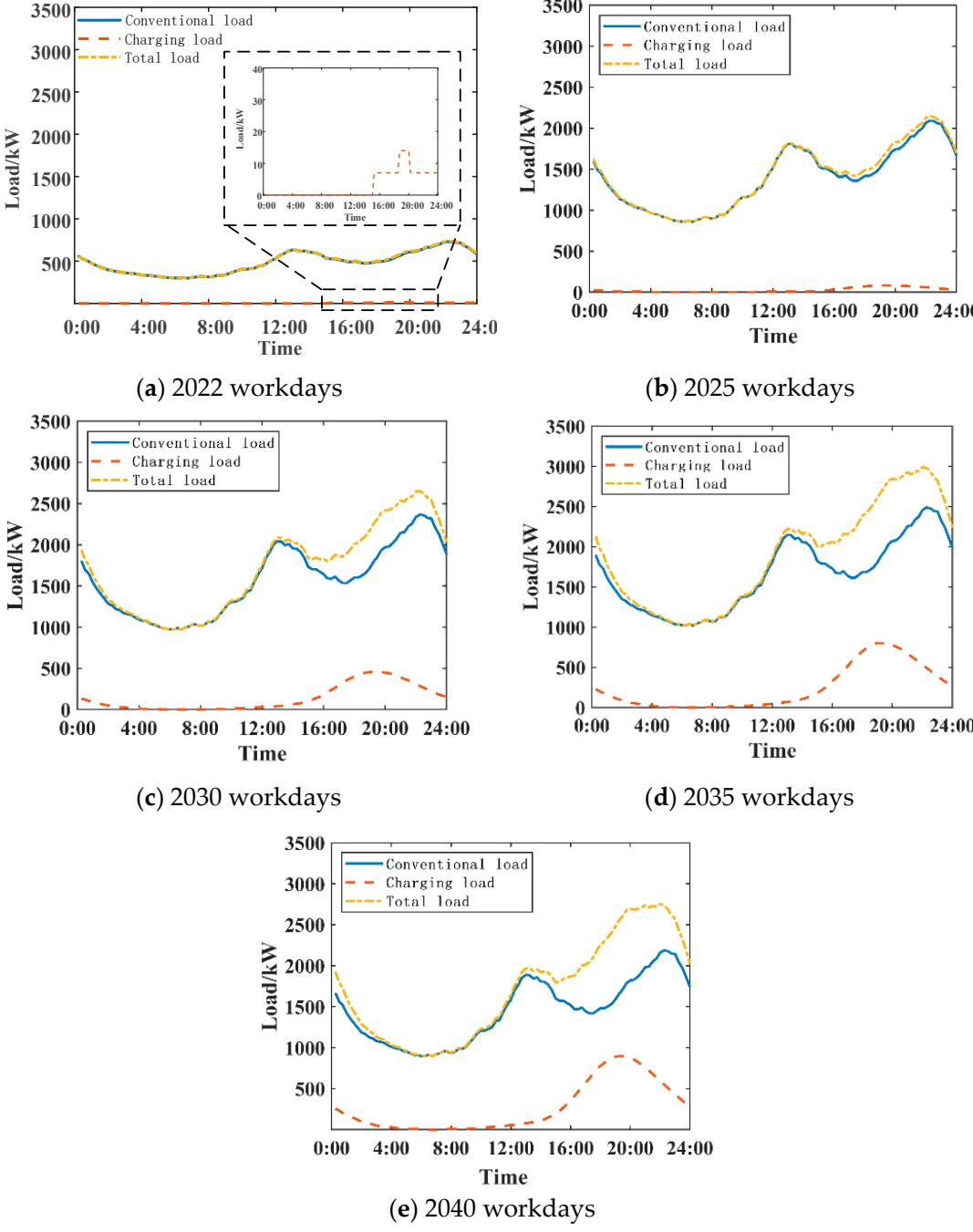

**Figure 11.** Load forecast for community areas.

### 6.3. Planning Scheme of Charging Facilities in Community

The charging habit of EV owners in the community is generally to charge on the same day and travel the next day. Therefore, the construction of EV charging piles in communities should be planned according to the number of charging vehicles needed daily, so as to meet the daily charging demand of EV users. Due to the influence of EV charging simultaneity rate in communities, the overall load of the charging node can be planned and constructed according to the daily charging load peak. The planning of charging nodes in the newly built communities based on the above planning ideas is shown in Table 2. The planning of one charging node is shown in the table, and the planning of the other node is the same.

**Table 2.** Planning for the construction of charging nodes in community areas.

| Year | Number of Charging Facilities to Build/Unit | Peak Load (Weekdays)/kW | Node Capacity Configuration/kVA |
| --- | --- | --- | --- |
| 2022 | 2 | 14 | 315 |
| 2023 | 5 | 21 | 315 |
| 2024 | 8 | 35 | 315 |
| 2025 | 12 | 49 | 315 |
| 2026 | 18 | 70 | 315 |
| 2027 | 26 | 98 | 315 |
| 2028 | 35 | 133 | 315 |
| 2029 | 48 | 182 | 315 |
| 2030 | 62 | 238 | 315 |
| 2031 | 75 | 287 | 630 |
| 2032 | 86 | 322 | 630 |
| 2033 | 95 | 357 | 630 |
| 2034 | 102 | 385 | 630 |
| 2035 | 107 | 406 | 630 |
| 2036 | 111 | 420 | 630 |
| 2037 | 114 | 427 | 630 |
| 2038 | 117 | 441 | 630 |
| 2039 | 119 | 448 | 630 |
| 2040 | 119 | 448 | 630 |

Using 300 m * 300 m as the power supply grid, the peak load density of the community in the saturation year (2040) is calculated. The grid division is shown in Figure 12. The load nodes at the boundary of the grid belong to the grid in the principle of going right and up. The calculation results of peak grid load density in this community are shown in Table 3.

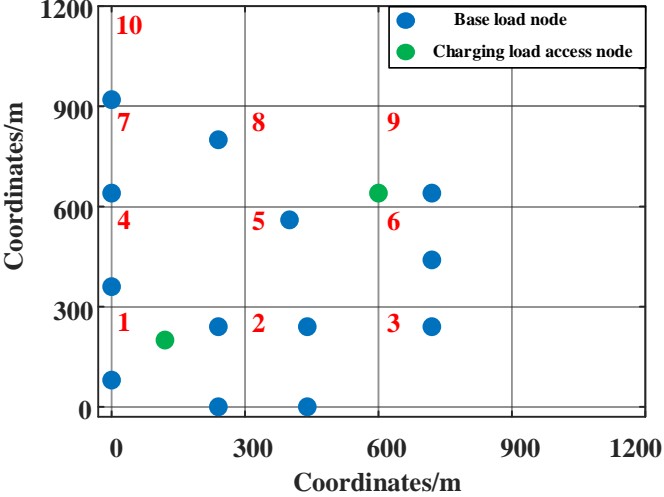

**Figure 12.** Community area power supply grid division.

**Table 3.** Peak load density of the power supply grid in the saturation year (2040) for community areas.

| Grid Number | Load Density (kW/km$^2$) | Grid Number | Load Density (kW/km$^2$) |
|---|---|---|---|
| 1 | 10,419.44 | 6 | 2563.33 |
| 2 | 1687.00 | 7 | 4781.78 |
| 3 | 783.22 | 8 | 0.00 |
| 4 | 3573.89 | 9 | 7113.89 |
| 5 | 2563.33 | 10 | 2136.11 |

According to the load density calculation results and the actual 10 kV line planning. 10 kV bus outlet nodes should be selected near grids with high load density. Therefore, grid 1 is selected as the grid where the 10 kV bus outlet node of the community distribution network is located, and the coordinates are set as (0, 0).

The peak base load data, charging load data, node information and line parameters of the community in saturation year were substituted into the collaborative optimization construction model of the community. The line construction cost and the planned service life of the line were set out in Table A4 of Appendix A. The initial population N was 100, the number of iterations was 100, the shift probability was 0.7, and the reallocation probability was 0.3. The Prim algorithm and single-parent genetic algorithm based on tree structure coding are used to solve the mathematical model of collaborative optimization construction. The optimal economic topology is solved in Figure 13, and the convergence process is shown in Figure 14. The specific construction data are in Table A5 in Appendix A.

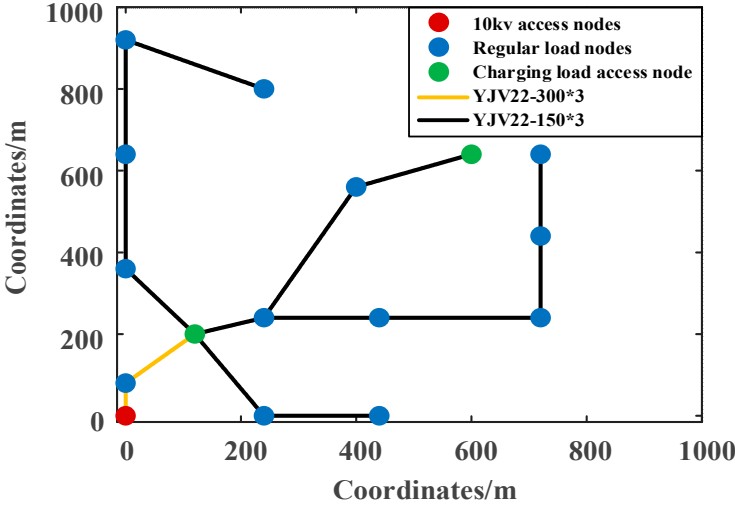

**Figure 13.** Community area line topology.

### 6.4. Summary

Taking the actual community as an example, firstly, combined with the occupancy rate and inventory prediction model, the community electric vehicle inventory is predicted to provide data support for subsequent charging facility planning. Through the basic power load forecasting model, the peak value of the basic load of each node is calculated in the community, and the required capacity of each node is planned at one time. By combining the charging habits of electric vehicle owners, charging at a simultaneous rate and charging peak load, the charging facilities and node capacity configuration are planned. According to the load density calculation results and the actual 10 kV line planning, grid 1 is selected as the grid where the 10 kV bus outlet nodes of the community distribution network are located. In this paper, the shortest path of community power supply is obtained from the Prim algorithm, and the optimal power supply topology is obtained by the single-parent genetic algorithm to minimize the construction and operation costs of the distribution network. This way can approach the optimal power supply topology before iterations,

and thus, can quickly converge when solving the optimal topology. It can be seen from Tables A4 and A5 that the main cost of the community distribution network comes from the line construction. In addition, because one of the charging load nodes is far away from the 10 kV line outlet node, and when the daily load is high, the network loss is relatively large and the operating cost is high.

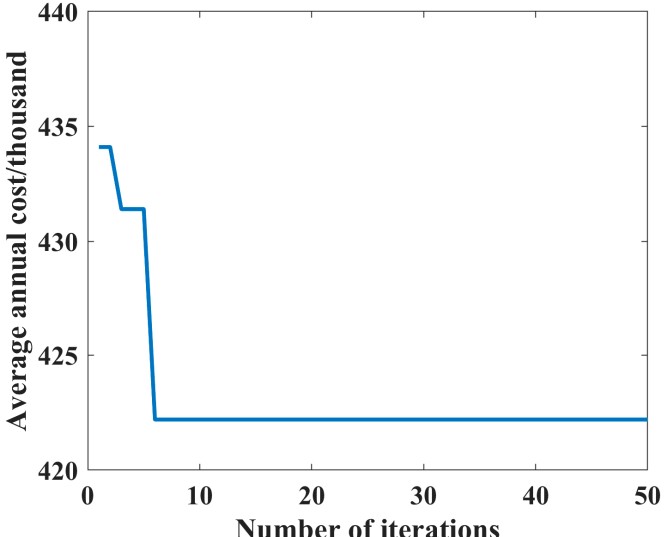

**Figure 14.** Convergence process.

## 7. Conclusions

Aiming at the problem of asset waste caused by the large planning margin of charging facilities in the community distribution network, this paper proposes a collaborative planning method for electric vehicle charging facilities and distribution networks in urban communities. The main contributions are as follows:

(1) Based on the load density method and the occupancy rate of the new community, the base load of the community is predicted; combined with the occupancy rate, the charging load of the community is predicted by the Bass model and the charging probability.

(2) According to the community charging loads and base loads, taking the minimum annual construction cost and operating cost of the community distribution network as the objective function, the Prim algorithm and the single-parent genetic algorithm are employed to optimize the power supply topology of the newly built community distribution network, and one-time planning of distribution network and annual construction planning scheme of charging facilities are given.

This method can be extended and applied to the construction of distribution networks and charging facilities in new urban communities. In this paper, the charging load forecasting is not comprehensive considering charging scenarios, and follow-up research will be further improved.

**Author Contributions:** Conceptualization, X.-H.D.; methodology, X.-H.D.; software, J.-W.J. and Z.-L.C.; validation, X.-H.D. and R.-Y.W.; data curation, J.-W.J.; writing—original draft preparation, J.-W.J. and J.Z.; writing review and editing, B.L. and X.Z. All authors have read and agreed to the published version of the manuscript.

**Funding:** Supported by Open Fund of Beijing Engineering Technology Research Center of Electric Vehicle Charging/Battery Swap(China Electric Power Research Institute) (No. YDB51202101508).

**Data Availability Statement:** The dataset used in this article can be obtained from the corresponding author under reasonable request.

**Conflicts of Interest:** The authors declare no conflict of interest.

## Appendix A

**Table A1.** Private vehicle ownership and growth rate in a city.

| Year | 2011 | 2012 | 2013 | 2014 | 2015 | 2016 | 2017 | 2018 | 2019 |
|---|---|---|---|---|---|---|---|---|---|
| Private car ownership/thousand | 1033 | 1277 | 1572 | 1933 | 2241 | 2564 | 2853 | 3158 | 3499 |
| Private car growth rate/% | 32.80 | 23.60 | 23.10 | 23.00 | 15.90 | 14.4 | 15.00 | 10.70 | 10.80 |

**Table A2.** Cable parameters and construction costs.

| Line Model | Cross-Sectional Area/mm$^2$ | Resistivity /Ω/km | Reactance /Ω/km | Ampacity /A | Line Construction Costs/Thousand/km |
|---|---|---|---|---|---|
| YJV22-3*150 | 150 | 0.12 | 0.103 | 358 | 900 |
| YJV22-3*300 | 300 | 0.305 | 0.089 | 540 | 1500 |

3*: three core cable.

**Table A3.** Peak base load and station capacity planning for communities in 2040.

| Number | Number of Households | Building Area/m$^2$ | Load Density W/m$^2$ | Area Requirement Factor | Annual Growth Rate | Average Daily Load/kW | Peak Load/kW | Capacity Configuration/kVA |
|---|---|---|---|---|---|---|---|---|
| 1# | 120 | 15,600 | 40 | 0.17 | 1% | 126.89 | 192.25 | 315 |
| 2# | 66 | 8580 | 40 | 0.17 | 1% | 69.79 | 105.74 | 160 |
| 3# | 44 | 5720 | 40 | 0.17 | 1% | 46.53 | 70.49 | 125 |
| 4# | 144 | 18,720 | 40 | 0.17 | 1% | 152.26 | 230.70 | 400 |
| 5# | 44 | 6600 | 40 | 0.17 | 1% | 53.68 | 81.34 | 125 |
| 6# | 44 | 5720 | 40 | 0.17 | 1% | 46.53 | 70.49 | 125 |
| 7# | 72 | 7920 | 40 | 0.17 | 1% | 64.42 | 97.61 | 160 |
| 8# | 72 | 7920 | 40 | 0.17 | 1% | 64.42 | 97.61 | 160 |
| 9# | 44 | 5720 | 40 | 0.17 | 1% | 46.53 | 70.49 | 125 |
| 10# | 120 | 15,600 | 40 | 0.17 | 1% | 126.89 | 192.25 | 315 |
| 11# | 180 | 27,000 | 40 | 0.17 | 1% | 219.61 | 332.75 | 500 |
| 12# | 174 | 26,100 | 40 | 0.17 | 1% | 212.29 | 321.65 | 500 |
| 13# | 174 | 26,100 | 40 | 0.17 | 1% | 212.29 | 321.65 | 500 |
| Total | | | | | | 1442.12 | 2185.04 | 3510 |

**Table A4.** Simulation parameters.

| Electrovalency/kWh | YJV22-3*150 Construction Costs/Thousand/km | YJV22-3*300 Construction Costs/Thousand/km | Operating Life/Years | Discount Rate | Maximum Load Operating Hours per Year/Hour |
|---|---|---|---|---|---|
| 0.5 | 900 | 1500 | 40 | 0.1 | 3000 |

3*: three core cable.

**Table A5.** Economic optimal topology construction data for communities.

| Average Annual Cost of Network Loss/Thousand | YJV22-3*150 Line Construction Length/m | YJV22-3*300 Line Construction Length/m | Average Annual Cost of Line Construction/Thousand |
|---|---|---|---|
| 104 | 3041.23 | 249.71 | 318.2 |

3*: three core cable.

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
