# Peer review of "Collaborative Planning of Community Charging Facilities and Distribution Networks"

_wevj, doi:10.3390/wevj14060143_

Round 1

Reviewer 1 Report

I have read with interest the article called "Coordinated Planning of New Residential Charging Facilities and Distribution Networks" I think the article is interesting and could find a place in the journal, although the problem is more of distribution (outside the vehicle) than in the vehicle.

The topics covered are many, starting from profiling to evaluating the positioning of power lines.

I really appreciated paragraph 2, which graphically introduces what was covered in the article. I would suggest using Figure 1 to guide the study of the following paragraphs. At the end of paragraph 2 I would put a short description to say that in paragraph 3 the assessment of electrical, residential and EV loads is discussed... In paragraph 4 ...

Prim's algorithm is well defined.

Paragraph 6 is very useful and interesting, I would say that it is the heart of the paper

I appreciated the use of the appendix to streamline the reading of the article.

Small tips:

1) check the references, often superscript characters are used in the square brackets

2) in the introduction I would avoid writing references [ ] but the authors in [ ]...

3) Figure 1 is very useful, I would use the descriptions in the figure for the names of the paragraphs

4) the description of formula 1 often has Is instead of is (also for other formulas)

5) check "The private car ownership and growth rate data of A city from 2011 to 2019 are shown in Table A1 of Appendix A"

6) put the space between number and unit of measure

7) better describes in table 2 what is meant by peak load (week-days)

8) I would increase the references, and check 2

Author Response

Comments:

Comment 1: check the references, often superscript characters are used in the square brackets

Response 1: All references in the article using superscript characters have been revised as required.

Comment 2: in the introduction I would avoid writing references [ ] but the authors in [ ].....

Response 2: Introduction have been revised as requested.

Comment 3: Figure 1 is very useful, I would use the descriptions in the figure for the names of the paragraphs

Response 3: The titles in the article all summarize the descriptions in Figure 1 as individual titles.

Comment 4: the description of formula 1 often has Is instead of is (also for other formulas)

Response 4: Change all references to "Is" in the article to "is"

Comment 5: check "The private car ownership and growth rate data of A city from 2011 to 2019 are shown in Table A1 of Appendix A"

Response 5: The data in Table A1 was checked again, and no errors were found.

Comment 6: put the space between number and unit of measure

Response 6: Leave a space between all numbers and units of measurement in the article.

Comment 7: better describes in table 2 what is meant by peak load (week-days)

Response 7: The peak charging load on weekdays is greater than that on weekends. In order to meet the charging needs of users, the distribution transformer capacity of charging nodes is configured according to the peak charging load on weekdays.

Comment 8: I would increase the references, and check 2

Response 8: Some references have been added to the introduction to enrich the article.

In addition, I have carefully revised some unclear and unreasonable places in the article. Finally, thank you very much for your guidance, I will keep working on it.The revised article has been added in the attachment, please check it. Thanks again!

Reviewer 2 Report

Please, see comments as pdf

The sentences tend to be lengthy and contain multiple ideas, which can make it challenging to follow the authors' arguments. Additionally, there are a few grammatical errors that could be addressed to improve clarity.

Author Response

Comment 1: General remarks: The abstract should be reworked to reflect the purpose and the findings of the

paper!

Specific remarks:

The first sentence is quite long and contains multiple ideas which can make it difficult to understand.

Additionally, there are a few grammatical errors that make it unclear what the writer is trying to convey.

A revised version of the sentence could be:

"When planning and constructing charging facilities in new a residential area, the supporting power grid

is often built based on expected saturation charging power predictions. This approach can result in large

amounts of idle distribution network resources in the early stages and waste of resources. To address

this problem, a method for predicting electric vehicle charging loads and planning charging facilities in

urban new residential areas is proposed."

The sentence “Then, taking the minimum annual average cost of distribution network in residential areas as the objective function, the power supply topology optimization model of the distribution network in new residential areas is established, and the Prim algorithm and single-parent genetic algorithm are explained. “ is long and complex, and that makes it difficult to understand the intended meaning. Also, once the model is established and the objective function is defined, it should be applied. In this case the sentence ends with the explanation of the algorithms.

A revised version of the sentence could be:

"To establish a power supply topology optimization model for the distribution network in new

residential areas, the objective function is set as the minimum annual average cost of the distribution

network in residential areas. The Prim algorithm and single-parent genetic algorithm are briefly

explained and afterwards applied."

Response 1: The abstract has been rewritten to more concisely express the purpose and results of the article's research.

Comment 2: Introduction

Consider a major reorganization of the related work and the background of the work.

The Introduction is not giving enough background of the research problem. Some of the references are

in Chinese only, for example [1, 17, 19 ...], some are outdated, some are not taken into consideration besides the similarity with the research topic, for example https://www.sciencedirect.com/science/article/pii/S2210670723001 841?via%3Dihub.

Response 2: Since there are not many articles in the new community, the introduction has been searched as much as possible, and the literature mentioned in the suggestion has been added to the introduction.

Comment 3: When citing a reference in a text it is better to say "in [x]" instead of "references [x]". The references are put in brackets [x] and not as a superscript [x].

Response 3: All references in the article have been revised as required.

Comment 4: When explaining what and how has been done, it is not enough just to explain Prim algorithm and single parent genetic algorithm but to successfully apply them.

Response 4: Added the passage to the introduction. “Considering that the topology is radial and satisfies the planned economy, this paper adopts the Prim minimum spanning tree algorithm to generate the initial topology of the distribution network, and uses the single-parent genetic algorithm to find the op-timal topology under different topologies. Finally, the method is verified by simulation with a practical case, and the plan of distribution network planning and charging facil-ities construction is given year by year.”

Comment 5: Specific remarks:

Therefore, the safe operation and reasonable planning of the distribution network are facing new

challenges [1]. - What are those challenges? [1] is not sufficient as a reference.

Repetition - As a result, the initial idle resources of the distribution network are large, causing a waste of

resources. – Particularly – which resources are wasted?

Although Bass model is well-known model in marketing and forecasting, it would be in order to use

reference for it. For example - Using the Bass model [X] to forecast the number of electric vehicles, [Y]

conducted a detailed analysis of the impact of the model's three main parameters (imitation coefficient,

innovation coefficient, and market potential) on the accuracy of the prediction.

Response 5: “the safe operation and reasonable planning of the distribution network are facing new Challenges”:

When a large number of electric vehicles are charged centrally at a certain time or place, the charging load agglomeration effect will be formed, resulting in an extreme peak of the load of the distribution network. As a result, the node voltage of the distribution network will exceed the limit and the network loss will increase, thus endangering the normal operation of capacitors, transformers and other equipment and damaging the service life of the equipment. Added relevant references.

which resources are wasted?” :

The number of charging facilities does not match the scale of electric vehicles, and the layout is unreasonable, etc.Added relevant references.

Issues related to Bass model references have been modified, and added relevant references.

Comment 6: Rest of the text, Sections 3 to 6

The acceptance capacity of the distribution network – use hosting capacity instead of acceptance.

Although Prim algorithm is a well-known minimum spanning tree algorithm that is widely used for a

variety of applications such as network design, routing, and clustering it should be referenced in the

text. There is no single explanation about this algorithm in the text. The single parent genetic algorithm

also needs a reference, although the idea of using only one parent to generate offspring in a genetic

algorithm is not new and has been explored in the literature. Therefore Fig. 6 and Fig.7 are unnecessary,

so do Fig. 4 and Fig. 5.

Response 6: In Section 5, Prim's Algorithm and Single parent genetic algorithm are briefly introduced, and relevant literature is cited. The irrelevant figures "Figures 4, 5, 6, 7" in the article were deleted.

Comment 7: Figure 8 is missing. Consider reorganization of the results presentation. Figure A2 and A3 are much more relevant to the body of the text then Fig. 15.

Response 7: Rearrange all the images in the article to fit them closer to the text.

Comment 8: At the end of Section 6 consider revision of the sentence – “Results The average annual cost of network loss mentioned is the cost under the scenario of planned saturation year. At this time, because one charging load node of the distribution network is far away from the 10kV outlet point and the conventional power load is high, the network loss cost is relatively large.”

Response 8: The conclusion of Section 6 of the article has been rewritten. Summarizes the work done in case analysis.

Comment 9: Conclusions

Consider revision of the Conclusions. It is not necessary to summarize what has been “mainly” done in the paper. Instead, a good conclusion should summarize the main points and contributions of the paper, and provide some context or implications for the findings. Therefore, it may be helpful to expand on the conclusion by providing additional information such as:

ï‚·1) Why is EV charging load forecasting and charging facility planning important in newly built residential areas?

ï‚·2) What are the key findings or results of the paper?

ï‚·3) What are the potential implications or applications of the paper's findings?

ï‚·4) What are some of the limitations or areas for future research?

By including this additional information, the conclusion can provide a more complete summary of the paper's contributions and help readers better understand its significance.

Response 9: Revised the conclusions. Summarizes the main viewpoints and contributions of the paper and briefly describes some limitations of the paper.

In addition, I have carefully revised some unclear and unreasonable places in the article. Finally, thank you very much for your guidance, I will keep working on it.The revised article has been added in the attachment, please check it. Thanks again!

Reviewer 3 Report

A very topical topic. A scientific work, written correctly. Congratulations to the authors.

Author Response

Thank you very much for your opinion, I will keep working on it.

Reviewer 4 Report

In the submitted paper, the Authors mainly introduces a forecasting method for the EV charging load and a planning approach for the charging facility for a residential area. More in depth a load density method and occupancy rate approach is adopted based on the prediction of EV charging load, combined with the relevant prediction of the occupancy rate.  The power supply topology optimization model of distribution network is used with the objective of minimizing the average annual cost of distribution network A genetic algorithm- based solution method is also proposed.

The paper is quite interesting, even is it should be presented in a more clear way, more details on the methodology need to be clearly discussed. Please find below my main concerns.

In the propose method, the role of the distribution network is not discussed clearly. How its limitations are included in the proposed approach? Please consider this point.

In (8), the right side of equation is not minF. It should be F. Consider to use a more clear formalization of the problem.

Please consider to clarify the prediction method adopted in the manuscript. It is suggested to clarify objective functions and constraints of the proposed optimization problem. Clarify also, in a summary inputs, outputs and optimization variables.

In Section V, please clarify the way the variables of the problem under study and discussed in the previous sections are implemented.

In the numerical application, the quality of figures should be improved. Also, units should be clearly reported in the axes.

The last figure shows that the convergence of the method is reached after few iterations. Is it normal? Is it sure that the obtained results are accurate. I think it is required to compare the proposed solution method with other available in the literature with respect to the application proposed in the last part of the paper.

Advantages of the proposed approach and its limitations should be clearly highlighted in the paper. Also, they should be discussed also in the Conclusion sections.

Moderate editing of English language

Author Response

Comment 1: In the propose method, the role of the distribution network is not discussed clearly. How its limitations are included in the proposed approach? Please consider this point.

Response 1: Due to the large idleness of the distribution network in the initial stage, serious asset waste is caused. To solve this problem, this paper proposes a collaborative planning method for urban community charging facilities and distribution network. And the community charging facilities and distribution network are planned year by year.

Comment 2: In (8), the right side of equation is not minF. It should be F. Consider to use a more clear formalization of the problem.

Response 2: In formula (8), minF is changed to F.

Comment 3: Please consider to clarify the prediction method adopted in the manuscript. It is suggested to clarify objective functions and constraints of the proposed optimization problem. Clarify also, in a summary inputs, outputs and optimization variables.

Response 3: First, based on the load density method and occupancy rate to predict the base electricity load in the community, the Bass model and charging probability are used to predict the community's electric vehicle charging load. Taking the minimum annual construction and operation costs of the community distribution network as the objective function, the power supply topology of distribution network for a new community is optimized by using Prim and single-parent genetic algorithms. Finally, the proposed scheme is veri-fied by using the actual community data of a certain city in China as an analysis example, and the scheme of one-time planning of distribution network and yearly construction of charging facilities is given.

Comment 4: In Section V, please clarify the way the variables of the problem under study and discussed in the previous sections are implemented.

Response 4: Because in Section 6, the actual calculation of how the problem variables studied and discussed in the previous sections are realized, there is no answer in Section 5.

Comment 5: In the numerical application, the quality of figures should be improved. Also, units should be clearly reported in the axes.

Response 5: Further improvements have been made to the quality of graphics in articles.

Comment 6: The last figure shows that the convergence of the method is reached after few iterations. Is it normal? Is it sure that the obtained results are accurate. I think it is required to compare the proposed solution method with other available in the literature with respect to the application proposed in the last part of the paper.

Response 6: The method does reach convergence after several iterations. There is little literature on the collaborative planning of new community charging facilities and distribution networks, so no comparison is made. The planning scheme proposed in the article may be only a part and needs further improvement.

Comment 7: Advantages of the proposed approach and its limitations should be clearly highlighted in the paper. Also, they should be discussed also in the Conclusion sections.

Response 7: Revised the conclusions. Summarizes the main viewpoints and contributions of the paper and briefly describes some limitations of the paper.

In addition, I have carefully revised some unclear and unreasonable places in the article. Finally, thank you very much for your guidance, I will keep working on it.The revised article has been added in the attachment, please check it. Thanks again!

Round 2

Reviewer 4 Report

I'm satisfied of the replies and modifications.

Quality of English Language is ok.